# The Genetic Background of the Immunological and Inflammatory Aspects of Progressive Supranuclear Palsy

**DOI:** 10.3390/ijms26093927

**Published:** 2025-04-22

**Authors:** Piotr Alster, Natalia Madetko-Alster

**Affiliations:** Department of Neurology, Medical University of Warsaw, Kondratowicza 8, 03-242 Warsaw, Poland

**Keywords:** PSP, MAPT, microglia, genetics, atypical parkinsonism

## Abstract

Progressive supranuclear palsy (PSP) is a neurodegenerative disease, classified as an atypical Parkinsonian syndrome, that has been pathologically and clinically defined. The histopathological aspects of the disease include tufted astrocytes, while the clinical features involve oculomotor dysfunction, postural instability, akinesia, cognitive impairment, and language difficulties. Although PSP is generally considered a sporadic disease, interest is growing in its genetics, with contemporary research focusing on familial backgrounds and neuroinflammation. Indeed, microglial activation and other inflammatory mechanisms of PSP pathogenesis have been extensively analyzed using genetic examinations to identify the factors impacting neurodegeneration. As such, this review aims to elaborate on recent findings in this field.

## 1. Introduction

Progressive supranuclear palsy (PSP) is a rare neurodegenerative disease characterized by four-repeat tauopathy [1]. Among its crucial pathological elements are tufted astrocytes, with clinical manifestations, including akinesia, postural instability, oculomotor dysfunction, cognitive impairment, and language difficulties [2]. The disease, which has a prevalence of 5.8 to 6.5 per 100,000 [3], is defined within specific domains that distinguish multiple clinical phenotypes. PSP–Richardson’s syndrome (PSP-RS) is the most common subtype and features pronounced oculomotor dysfunction, an early presence of falls, and a lack of responsiveness to levodopa. PSP–parkinsonism-predominant (PSP-P) is an entity that, due to its clinical overlap with Parkinson’s disease (PD), may be indistinguishable in the early stages [4]. Among the less common PSP subtypes are PSP–pure akinesia with gait freezing (PSP-PAGF), PSP–corticobasal syndrome (PSP-CBS), PSP–behavioral variant of frontotemporal dementia (PSP-bvFTD), PSP–progressive non-fluent aphasia (PSP-PNFA), PSP–frontal-dominant, PSP with predominant cerebellar ataxia, and PSP with oculomotor dysfunction [2].

PSP is characterized by pronounced clinical deterioration; inefficient diagnosis through in vivo diagnostic tools such as magnetic resonance imaging (MRI), positron emission tomography (PET), and single-photon emission computed tomography (SPECT); and a lack of evidence-based treatments. Indeed, in vivo examination only enables a possible or probable diagnosis, with a definite diagnosis dependent on neuropathological evaluation [2]. PSP is generally considered a sporadic disease, although research into a link between familial genetics and increased risk of the disease is evolving alongside investigations into genetic aspects of PSP and inflammation-induced neurodegeneration. Despite multiple studies on the pathogenesis of PSP indicating a possible impact of glial activation, it is unclear whether inflammation is the cause or consequence of neurodegenerative mechanisms [5,6]. In PSP, the issue of autoimmune mechanisms was brought up based on HLA phenotyping in neuropathologically verified PSP cases [7]. Wang et al., in a preliminary study, showed that epitopes within the tau peptide may have affinity to HLA alleles, which could be detected in a subset of PSP patients. Anti-IGLON5, a disease that clinically partially overlaps with PSP, is associated with the presence of HLA-DRB1*10:01 and HLA-DQB1*05:01 alleles [8]. Certain features of microglial activation and cytokines derived therefrom may be linked with a potentially beneficial impact on clinical impairments, contrary to their role in atrophic changes [5,6,9,10]. Additionally, different clinical subtypes are associated with diverse interleukin and inflammatory factor profiles that, considering the impact of tau loading and its distribution, may be determined by genetics [11,12,13,14]. Therefore, the current review provides a perspective on the features linking the genetic and inflammatory aspects of PSP.

## 2. The Genetic Background of Progressive Supranuclear Palsy—A General Overview

A genome-wide association study (GWAS) by Farrell et al. revealed that factors such as cortical expression of syntaxin 6 (*STX6*), RUNX family transcription factor 2 (*RUNX2*), and myelin-associated oligodendrocyte basic protein (*MOBP*) are linked with increased PSP risk [15]. A transcriptome-wide association analysis in the same study stressed the significance of multiple genes in the *6p21.32* and *17q21.32* loci, including complement C4A (*C4A*), complement C4B (*C4B*), and the microtubule-associated protein tau gene (*MAPT*). As such, the research indicates the existence of cell-specific effects on gene expression in oligodendrocytes.

The PSP tau pathology is linked to ubiquitin-fold modifier-1 (UFM1) substrate modification (UFMylation) through cascade disruption in neurofibrillary-tangle-bearing neurons [16]. Fujioka et al. analyzed 19 descriptions of familial PSP and identified Parkinsonism and oculomotor dysfunction as the most common clinical aspects of familial PSP [17]. Mutations in *MAPT* are considered the most frequent factor associated with familial PSP, while deviations linked with PSP are observed in the *17q21.31* locus and are associated with low copy number repeats accompanied by genomic inversion [18] (Table 1).

Patients with *MAPT* mutations are affected by the earlier initiation of PSP symptoms than carriers of less common genetic features such as leucine-rich repeat kinase 2 (*LRRK2*) or dynactin subunit 1 (*DCTN1*) [19]. *MAPT* variants are also linked with frontotemporal dementia (FTD). In particular, the mutation N279K associated with region Ex10 is associated with FTDP-17. Furthermore, whole-exome sequencing in a 61-year-old female patient with a clinical manifestation of PSP revealed the E342K variant of *MAPT* [20], while the homozygosity linked with the *MAPT* H1 haplotype is considered to be a risk factor for PSP and corticobasal degeneration (CBD) [21]. As such, multiple theories exist around the overlapping of neurodegeneration in both tauopathies [22]. *MAPT* mutations (*MAPT c.1024G>A*, *p. Glu342Lys*, and *MAPT c.1217 G>A*, *p. Arg406Gln*) were detected among patients with PSP-RS who developed features commonly associated with synucleinopathies, including autonomic failure and dream enactment behavior [23]. Similarly, as in Alzheimer’s disease (AD), PSP pathogenesis involves the joint expression of specific N- and C-terminal *MAPT* isoforms [24].

Recent studies indicated the possible significance of PSP genetic risk factors. An examination of the *17q21.31* chromosome region and extended evaluations of α, β, and γ whole-genome duplications found an association between γ duplications and PSP risk [25].

The characteristic classical PSP phenotype symptoms may be due to entirely different entities mimicking a PSP clinical manifestation. In a case using SPECT to examine a patient with clinical features of the disease, investigators observed pronounced atrophic changes within the midbrain combined with dopamine transporter uptake deficiencies [26]. Meanwhile, a genetic evaluation revealed a novel in-frame deletion/insertion mutation in exon 3 of glial acidic fibrillary protein (*GFAP*).

PSP may be linked with pathogenic sphingomyelin phosphodiesterase 1 (*SMPD1*) variants associated with Niemann–Pick disease types A and B as its recessive cause. A study by Lim et al. presented three unrelated patients affected by the heterogenous form of the gene and a clinical manifestation of PSP-RS [27].

An evaluation of bassoon presynaptic cytomatrix protein (*BSN*) in Japanese patients with neuropathologically verified PSP highlighted two mutations at *p.Thr2542Met* and *p.Glu2759Gly*, with the former associated with cognitive impairment, Parkinsonism, and behavioral changes. On the other hand, the *p.Glu2759Gly* mutation was linked with cognitive impairment, right-dominant motor impairment, right limping gait, and postural instability [28]. The mutations differed in the context of atrophic changes, as *p.Thr2542Met* was associated with hippocampal atrophy and *p.Glu2759Gly* with left-dominant parietal atrophy. Nonetheless, atrophic changes in the midbrain tegmentum were observed in both cases. Interestingly, an analysis of neurofibrillary tangle occurrence revealed different disseminations, with tangles observed in the globus pallidus, subthalamic nucleus, substantia nigra, and CA1 in the first case and the globus pallidus, subthalamic nucleus, substantia nigra, thalamus, putamen, and brainstem tegmentum in the second mutation. However, any interpretation of the general significance of these mutations is limited since the study involved only two cases.

Whole-genome sequencing by Wang et al. found—alongside the genetic loci commonly linked with PSP pathophysiology, such as *MAPT*, *MOBP*, *STX6*, solute carrier organic anion transporter family member 1A2 (*SLCO1A2*), dual-specificity phosphatase 10 (*DUSP10*), and SP1 transcription factor (*SP1*)—signals for apolipoprotein E (*APOE*), FCH and mu domain-containing endocytic adapter 1 (*FCHO1*)/*MAP1S*, *kinesin family member 13A* (*KIF13A*), *tripartite motif-containing 24* (*TRIM24*), *tenascin XB* (*TNXB*), and *ELVOL fatty acid elongase 1* (*ELOVL1*). Interestingly, *APOE* ε2, an allele with a protective role in AD pathogenesis, was linked with increased PSP risk [29,30,31]. Such discrepancies may be crucial in the context of recently described differences between the tau-mediated inflammatory processes of *APOEε2* and *APOEε4* astrocytes in these two diseases [30,31,32]. Other studies indicate that the association between *APOE* and PSP risk factors is more ambiguous in the non-Caucasian population [33].

## 3. Immune Molecular Pathogenesis in Progressive Supranuclear Palsy

Although the PSP pathomechanism pathways remain unclear, hypotheses focusing on immune activation [34] are supported by evidence of the tufted astrocyte pathology being significantly enriched in microglial genes [35] and abnormal microglial transcript upregulation associated with enhanced astrocytic tau. In this context, the outcome of a GWAS by Farrell et al. on the colocalization of abnormal tau species in oligodendrocytes with *C4A* protein, using a novel pattern in deoxyribonucleic acid in PSP analysis, highlights the potential significance of internal immune functioning [15]. Neurodegeneration pathways leading to tauopathies are linked by factors that, contrarily, activate and suppress microglial activation [34]. Rexach et al. employed a weighted gene co-expression analysis to evaluate transgenic models with human mutant *MAPT* and identify co-expressed gene modules in microglia and brain tissue [34]. Their study found 13 co-expressed modules, with 7 containing microglial genes. The modules were sorted according to distinct trajectories with respect to disease progression, including (1) the earliest stages before neuronal loss and present through later stages (M_UP1, M_DOWN1, and M_DOWN2), (2) early periods of neuronal loss (M_UP2 and M_DOWN3), and (3) late-stage neurodegeneration (M_UP3) [34]. The disease periods differed in the context of genes impacting pro-inflammatory (M_UP1) and anti-inflammatory (M_DOWN1) mechanisms, leading to homeostasis deviations. Interestingly, the microglial activation patterns differed between PSP and other tauopathies, such as AD and FTD–Pick’s disease. In this context, the evaluation of modules in the area of microglial activation could explain abnormalities in the levels of microglial-derived factors such as interleukin 1-beta, interleukin 6, and interferon gamma [12]. For genetic risk, PSP was matched with EARLY_DOWN1, FTD–Pick’s disease with EARLY-DOWN1 and EARLY-UP1, and AD with M_UP3. The study’s authors demonstrated a dual-element process involving primary suppression through type II interferon with subsequent pro-inflammatory mechanisms [34]. Follow-up research by Rexach et al. based on the use of snRNAseq and snATACseq showed that PSP astrocytes have high chromatin accessibility in normally hypermethylated regions, indicating de-repressed gene expression [36].

An analysis of module eigengenes showed their varying impact on PSP’s pathophysiology, with higher expression of M4, M6, and M12 linked to the disease [37]. Cell enrichment in module M4 was linked with astrocyte genes, and M6 was linked to microglial/endothelial genes. Meanwhile, M3 was associated with oligodendroglia and lower expression in PSP, and M6 coincided with differentially expressed chemokine ligand 3 (*CL3*). An issue with microglial genes was also raised in a study evaluating ABI family member 3 (*ABI3*) and phospholipase C gamma 2 (*PLCG2*) missense variants as neurodegenerative disease risk factors among Caucasians and African Americans. Although the study was primarily based on AD assessment, it included 231 PSP patients [38] and found that microglial gene-enriched co-expression networks increased significantly in AD but not in PSP. Another study showed an association between amyloid beta, but not tau, and microgliosis, microglial activation, and increased *PLCG2* levels in the brain [39]. However, the study showed a link between *PLCG2_rs72824905-G* and tau neuropathology suppression in PSP. Furthermore, *rs72824905-G* was associated with longevity and indicated as a factor in reducing the risk of AD, dementia with Lewy bodies, and FTD but not that of PD [40].

Triggering receptor expressed in myeloid cells 2 (*TREM2*) is generally connected to the microglial-driven phagocytosis of apoptotic neuronal cells, though abnormalities in this gene can be associated with the pathogenesis of neurodegenerative disorders [41]. Indeed, an assessment of microglia-related *TREM2* revealed upregulation in the substantia nigra when 24 neuropathologically confirmed cases of PSP were compared with 14 controls [42]. Additionally, *TREM2* messenger ribonucleic acid (mRNA) corresponded with hyperphosphorylated neuronal tau in the substantia nigra. However, an analysis of *TREM2* in tauopathic mouse models showed an ambiguous role of this gene, with abnormal tau enhancement and accumulation found [43]. *TREM2* variants were also linked with atypical Parkinsonism in multiple pathologies, including synucleinopathies in the context of multiple system atrophy [44]. *TREM2* was also evaluated in PD and FTD, with further assessment revealing that *TREM2* p.R47H substitution can be interpreted as a risk factor for AD, PD, and FTD [41]. On the other hand, an analysis of the frequency of the *TREM2* p.R47H variant using a large cohort comprising AD, FTD, mild cognitive impairment, PSP, CBS, and amyotrophic lateral sclerosis (ALS) cases and controls did not find any significant links between the variant and neurodegenerative diseases other than AD [45]. Carrying p.R47H in the TREM2 gene was associated with mild cognitive impairment, amyloid deposition, and microglial activation in a study on influenza vaccine immune challenge [46]. This variant was found to show ambiguous results in the context of the risk of neurodegenerative diseases. It was indicated as a possible risk factor of sporadic amyotrophic lateral sclerosis (ALS) and essential tremor [47,48]. Contrary studies based on the examination of FTD, PD, and ALS did not show results indicating p.R47H as a definitive risk factor of these diseases [49,50]. In AD, *TREM2* mutations were linked with phosphorylated tau (p-tau)181 and soluble TREM2 and were interpreted as being related to an unfavorable clinical course. Such an analysis has yet to be undertaken for PSP [51].

PSP and CBD involve early/immediate inflammatory responses that result in a p-tau signature [52]. Briel et al. showed that, apart from those for transcription factor EB (TFEB) and nuclear factor IC (NFIC)/T-cell leukemia homeobox 1 (TLX1), the transcription factor site accessibilities are similar in both diseases. The crucial discrepancies are associated with the extent of accessibility alterations, with Jun proto-oncogene (*JUN*) (*B*, *D*)*_HIGH_*, Fos proto-oncogene (*FOS*) (*L1*, *L2*)*_HIGH_*, double homeobox 4 (*DUX4*), KLF transcription factor 5 (*KLF5*), MYC-associated factor X (*MAX*)/MYC proto-oncogene (*MYC*), paired box 6 (*PAX6*), and peroxisome proliferator-activated receptor gamma (*PPARG*) suggesting CBD astrocytes, and decreased *TFEB* and cAMP responsive element-binding protein 1 (*CREB1*) accessibilities indicative of PSP astrocytes [52]. The genetic risk variants of PSP and FTD were mapped to astrocytic chromatin accessibility profiles, suggesting a complex (epi-)genetic link to astrocytic pathology, with, notably, no enrichments in microglial profiles [44].

The shared features of PSP, CBD, PD, and FTD pathogenesis have been described in the context of C-X-C chemokine receptor type 4 (*CXCR4*) [53,54,55], which is involved in microglial activation and increases the likelihood of developing these diseases. Increased *CXCR4* expression was detected in the ventral thalamus, subthalamus, pontine tegmentum, and cerebellar nuclei, which are partially linked with the structures identified in PSP staging [52,53,54,55,56]. Further analyses highlighted the significance of *CXCR4* in astroglial signaling abnormalities in neurodegenerative processes [55]. Additionally, *CXCR4* interacts with microglia-related cysteine-X-cysteine chemokine ligand 12 (*CXCL12*), toll-like receptor 2 (*TLR2*), RAS-like proto-oncogene B (*RALB*), and C-C motif chemokine receptor 5 (*CCR5*) [55]. Moreover, *CCR5* is associated with blood–brain barrier integrity [55], while *CXCR4* is indicated as impacting cognitive deterioration in neurodegenerative disease within an axis including *CXCL12* and *CXCR7*. Another study highlighted the possible impact of this axis on dopaminergic neuron survival, though it did not focus on PSP [57].

*MAPT* dysregulation is thought to affect the astrocyte tau distribution in PSP [11], with astrocytic tau buildup found prior to its accumulation in neurons. Experiments using a p-tau antibody (AT8) revealed that AT8+ astrocytes had significantly more *MAPT* transcripts and that deviations in endogenous astrocyte mRNA resulted in increased astrocytic tau accumulation in PSP. Furthermore, this process was not associated with AD. Other genetic variants linked with neuropathologically verified PSP include *STX6* and eukaryotic translation initiation factor 2 alpha kinase 3 (*EIF2AK3*) [57,58], with upregulated *STX6* expression found in the cerebellum [17].

## 4. Discussion

The associations between microglial activation and astrocytic tau in PSP and the evaluation of their molecular background raise questions about the causes of the different disease subtypes. To the best of our knowledge, no studies have been performed to genetically examine PSP in the context of its clinical subtypes. Based on the features of genes related to microglial activation, such as deviations in *TREM2* and their impact on cognitive deterioration and differences in cognition between PSP subtypes, it may be assumed that the diversity of these abnormalities plays a role in the clinical evolution of the disease [2,9,55]. However, the mechanism presented does not provide sufficient data to explain the pace of neurodegeneration in PSP compared to other neurodegenerative disorders. Moreover, the variability of immune processes leading to neurodegeneration in PSP raises concern over the repeatability of such mechanisms. Furthermore, the relatively low number of cases evaluated in genetic studies of immune processes in PSP limits an extended analysis of this issue. Nonetheless, some studies highlighting the genetic background in related conditions may be applicable to PSP due to the extent of overlap between such disorders.

## 5. Conclusions

The inflammatory mechanisms in PSP are an understudied issue in the analysis of the pathomechanisms of this disease (Figure 1). The diversity of paths leading to neurodegeneration in this process has been studied in terms of their causes and consequences. Inevitably, the genetic background of inflammation may partly explain the wider picture of clinical manifestations of PSP. It is not clear why certain cases develop into PSP-RS associated with pronounced clinical deterioration, while others are linked to more gradual evolution. Understanding PSP’s pathomechanisms may help to progress research on the treatment of this currently incurable disease, and it seems inevitable that evaluations of the genetic aspects of PSP will be crucial in furthering our understanding.

## Figures and Tables

**Figure 1 ijms-26-03927-f001:**
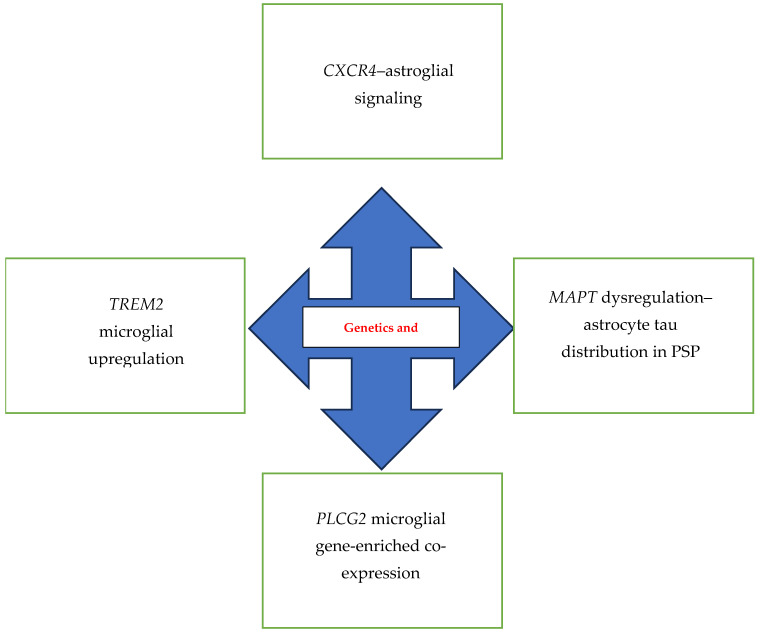
Aspects of immune molecular pathogenesis of progressive supranuclear palsy (PSP).

**Table 1 ijms-26-03927-t001:** Crucial features of *MAPT* in progressive supranuclear palsy (PSP).

	Pathogenesis of PSP
Comparison to other genetic factors	⇨ more frequently linked to the early onset of PSP than leucine-rich repeat kinase 2 (*LRRK2*) or dynactin subunit 1 (*DCTN1*)
Clinical implications	⇨ early onset of PSP ⇨ detected in patients with PSP–Richardson’s syndrome who developed features commonly associated with synucleinopathies, including autonomic failure and dream enactment behavior
Other features	*⇨* the *MAPT* H1 haplotype is considered to be a risk factor for PSP *⇨* impact on astrocyte tau distribution

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
