# Peer review of "The Genetic Background of the Immunological and Inflammatory Aspects of Progressive Supranuclear Palsy"

_ijms, 2025, doi:10.3390/ijms26093927_

Round 1
Reviewer 1 Report
Comments and Suggestions for Authors
This manuscript presents an interesting opinion statement about the role if immunological factors in PSP. The authors discuss the potential importance of genetic and other molecular levels that might etiologically be linked to PSP and neuroinflammation. However, several areas require clarification or improvement:
Specific comments:
- L34: Please replace "failure to respond to levodopa" with "lack of responsiveness to levodopa" as "failure" implies a voluntary process.
- L37f: I would recommend adhering to the 2017 MDS criteria and recognized PSP subtypes:
- Classical Richardson syndrome (PSP-RS), PSP-parkinsonism (PSP-P), PSP-pure akinesia with gait freezing (PSP-PAGF), PSP-corticobasal syndrome (PSP-CBS), PSP-behavioral variant of frontotemporal dementia (PSP-bvFTD), PSP-progressive non-fluent aphasia (PSP-PNFA)
- L47: Please refer to the more general term "glial activation" instead of specifying microglial or astroglial activation, as at this stage it is unclear to which extent both cell types contribute to the overall pathology.
- L65: Clarify if "tauopathic pathology" refers to tau pathology, or what this redundant phrase means.
- L72: Please specify whether you are referring to PSP with MAPT variants or FTDP-17 MAPT causal mutations with PSP phenotype.
- L74: Clarify which specific MAPT variants are meant, or reference the set of MAPT risk variants linked with FTD.
- L80: Please correct the typographical error "among in patients."
- L86: Briefly explain the meaning of "alpha, beta and gamma duplications."
- L88 and 94: Since Lim et al. did not confirm the diagnosis of PSP neuropathologically, please refer to the "classical PSP phenotype" rather than "PSP."
- L130: Specify what is meant by "novel genetic signal."
- L138-146: Describe the functional biological role of the modules rather than using study-specific module names. Mention specific inflammatory mechanisms (e.g., Interferon X activation).
- Additional consideration: Reference the follow-up work by Rexach et al. https://www.sciencedirect.com/science/article/abs/pii/S0092867424009103 using snRNAseq and snATACseq. showing that PSP astrocytes have high chromatin accessibility in normally hypermethylated regions, indicating de-repressed gene expression. This is important because gene patterns associated with microglial activation in bulk RNA-seq might be misattributed when there is aberrant gene regulation in astrocytes.
- L178: Change "Trem2" to "TREM2" and replace "deficient clinical course" with "unfavorable clinical course."
- L190: Note that in the cited study, genetic risk variants of PSP and FTD were mapped to astrocytic chromatin accessibility profiles, suggesting a complex (epi-)genetic link to astrocytic pathology, with notably no enrichments in microglial profiles.
- General comment: Include a brief mention of IgLON5 and its link with an acquired but genetically driven (HLA-association) tauopathy (3R/4R).
- The authors should consider including these preliminary results on HLA-phenotyping in autopsy-confirmed PSP cases, as it is immediately connected with the topic: https://www.biorxiv.org/content/10.1101/2025.01.14.632901v1
- General comment: When referring to FTD, please specify the underlying neuropathology (e.g., Pick's Disease) as FTD encompasses various biological entities.
Final Recommendation:
Major revision. The manuscript addresses an important topic but needs clarification of terminology and several adaptations with regard to the precise description of molecular findings in PSP.
Author Response
Dear Reviewer 1,
We would like to thank you for the opportunity to revise the manuscript and the referee’s for their valuable comments. We feel that the changes we made, based on their recommendations have improved the quality of our manuscript. The changes are highlighted in red. We would appreciate if you would now reconsider the manuscript for publication. Below authors provided responses to the comments accordingly.
Best regards
Piotr Alster and Natalia Madetko-Alster
- L34: Please replace "failure to respond to levodopa" with "lack of responsiveness to levodopa" as "failure" implies a voluntary process.
The change was implemented.
- L37f: I would recommend adhering to the 2017 MDS criteria and recognized PSP subtypes:
- Classical Richardson syndrome (PSP-RS), PSP-parkinsonism (PSP-P), PSP-pure akinesia with gait freezing (PSP-PAGF), PSP-corticobasal syndrome (PSP-CBS), PSP-behavioral variant of frontotemporal dementia (PSP-bvFTD), PSP-progressive non-fluent aphasia (PSP-PNFA)
The change concerning indicating the subtypes in accordance to MDS criteria was implemented.
- L47: Please refer to the more general term "glial activation" instead of specifying microglial or astroglial activation, as at this stage it is unclear to which extent both cell types contribute to the overall pathology.
The change was implemented.
- L65: Clarify if "tauopathic pathology" refers to tau pathology, or what this redundant phrase means.
Authors refer to tau pathology.
- L72: Please specify whether you are referring to PSP with MAPT variants or FTDP-17 MAPT causal mutations with PSP phenotype.
In this line authors refert to PSP with MAPT variants.
- L74: Clarify which specific MAPT variants are meant, or reference the set of MAPT risk variants linked with FTD.
Authors refered to mutation N279K in the region Ex10. Adequate information was added in the manuscript.
- L80: Please correct the typographical error "among in patients."
The correction was implemented.
- L86: Briefly explain the meaning of "alpha, beta and gamma duplications."
The mentioned duplications refer to whole genome resulting in two copies of each gene. The statement in the manuscript was rephrase with the acknowledgement of refering to whole genome duplication.
- L88 and 94: Since Lim et al. did not confirm the diagnosis of PSP neuropathologically, please refer to the "classical PSP phenotype" rather than "PSP."
The correction was implemented.
- L130: Specify what is meant by "novel genetic signal."
The statement was rephrased: In this context, the outcome of a GWAS study by Farrell et al. on the colocalization of abnormal tau species in oligodendrocytes with C4A protein, using a novel pattern in deoxyribonucleic acid in PSP analysis, highlights the potential significance of internal immune functioning [13].
- L138-146: Describe the functional biological role of the modules rather than using study-specific module names. Mention specific inflammatory mechanisms (e.g., Interferon X activation).
Information on biological significance was implemented.
- Additional consideration: Reference the follow-up work by Rexach et al. https://www.sciencedirect.com/science/article/abs/pii/S0092867424009103 using snRNAseq and snATACseq. showing that PSP astrocytes have high chromatin accessibility in normally hypermethylated regions, indicating de-repressed gene expression. This is important because gene patterns associated with microglial activation in bulk RNA-seq might be misattributed when there is aberrant gene regulation in astrocytes.
The information was implemented.
- L178: Change "Trem2" to "TREM2" and replace "deficient clinical course" with "unfavorable clinical course."
The change was implemented.
- L190: Note that in the cited study, genetic risk variants of PSP and FTD were mapped to astrocytic chromatin accessibility profiles, suggesting a complex (epi-)genetic link to astrocytic pathology, with notably no enrichments in microglial profiles.
The important statement suggested by the reviewer was added accordingly.
- General comment: Include a brief mention of IgLON5 and its link with an acquired but genetically driven (HLA-association) tauopathy (3R/4R).
The short acknowledgement of anti-IGLON5 was implemented in the introduction.
- The authors should consider including these preliminary results on HLA-phenotyping in autopsy-confirmed PSP cases, as it is immediately connected with the topic: https://www.biorxiv.org/content/10.1101/2025.01.14.632901v1
Authors implemented this issue in the introduction refering to published research Wang J, Forrest SL, Dasari S, et al. Investigation of the HLA locus in autopsy-confirmed progressive supranuclear palsy. Immunobiology. Published online March 22, 2025. doi:10.1016/j.imbio.2025.152892
- General comment: When referring to FTD, please specify the underlying neuropathology (e.g., Pick's Disease) as FTD encompasses various biological entities.
Authors fully agree with this comment. In the study in which the neuropathology was indicated, adequate information was implemented, however multiple studies lacked information on this issue.
Final Recommendation:
Major revision. The manuscript addresses an important topic but needs clarification of terminology and several adaptations with regard to the precise description of molecular findings in PSP.
The modifications were performer according to the reviewer’s suggestions.
Reviewer 2 Report
Comments and Suggestions for Authors
The authors sent for publication an Opinion article entitled “Genetic background of the immunological and inflammatory aspects of progressive supranuclear palsy”. The authors discussed current knowledge and several topics related to the pathogenesis, neuropathology and genetic aspects associated with Progressive Supranuclear Palsy (PSP). This is an interesting manuscript, which could be even improved in its quality considering some points for discussion:
- I suggest authors to include a figure or a diagram with a scheme showing the most important aspects related to the immune molecular pathogenesis and the genetic basis of PSP.
- Carriers of the variant p.R47H in the TREM2 gene have been correlated with the occurrence of both tau and amyloid depoisition, microglial activation, and mild cognitive impairment and Alzheimer’s disease (J Neuroinflammation 2023;20(1):272). One previous study has demonstrated this variant as risk factor for sporadic ALS (JAMA Neurol 2014;71(4):449-53). Other study showed no definite role of this variant as a risk factor for Frontotemporal Dementia (Neurobiol Aging 2014;35(2):444.e1-4). Other studies did not show any potential risk factor for FTD, PD and ALS (Alzheimers Dement 2015;11(12):1407-16). Other study disclosed that the variant could have a potential role as a risk factor for essential tremor (Parkinsonism Relat Disord 2015;21(3):306-9). Other study failed to show association of this variant with ALS (Mol Biol Rep 2021;48(3):2601-10).
- The description of the p.R47H variant should be corrected in lines 173 and 174 and italics must be removed.
- The manuscript does not include a “Conclusion” topic, which should be added.
Author Response
Dear Reviewer 2,
We would like to thank you for the opportunity to revise the manuscript and the referee’s for their valuable comments. We feel that the changes we made, based on their recommendations have improved the quality of our manuscript. The changes are highlighted in red. We would appreciate if you would now reconsider the manuscript for publication. Below authors provided responses to the comments accordingly.
Best regards
Piotr Alster and Natalia Madetko-Alster
- I suggest authors to include a figure or a diagram with a scheme showing the most important aspects related to the immune molecular pathogenesis and the genetic basis of PSP.
A figure was added according to the reviewer’s suggestions.
- Carriers of the variant p.R47H in the TREM2 gene have been correlated with the occurrence of both tau and amyloid depoisition, microglial activation, and mild cognitive impairment and Alzheimer’s disease (J Neuroinflammation 2023;20(1):272). One previous study has demonstrated this variant as risk factor for sporadic ALS (JAMA Neurol 2014;71(4):449-53). Other study showed no definite role of this variant as a risk factor for Frontotemporal Dementia (Neurobiol Aging 2014;35(2):444.e1-4). Other studies did not show any potential risk factor for FTD, PD and ALS (Alzheimers Dement 2015;11(12):1407-16). Other study disclosed that the variant could have a potential role as a risk factor for essential tremor (Parkinsonism Relat Disord 2015;21(3):306-9). Other study failed to show association of this variant with ALS (Mol Biol Rep 2021;48(3):2601-10).
Authors are grateful for this point, which was implemented in the part concerning variant p.R47H.
- The description of the p.R47H variant should be corrected in lines 173 and 174 and italics must be removed.
The change was implemented.
- The manuscript does not include a “Conclusion” topic, which should be added.
The conclusion section was implemented.
Round 2
Reviewer 1 Report
Comments and Suggestions for Authors
The authors addressed the comments adequately.